# Absence of posterior pituitary bright spot in adults with CNS tuberculosis: A case-control study

**Smitesh G. G.**[1], **Pavithra Mannam**[2], **Vignesh Kumar**[1], **Tina George**[1]*, **Murugabharathy K.**[1‡], **Turaka Vijay Prakash**[3‡], **Bijesh Yadav**[4], **Thambu David Sudarsanam**[1]

**1** Department of General Medicine, Christian Medical College Vellore, Vellore, Tamil Nadu, India,
**2** Department of Radiology, Christian Medical College Vellore, Vellore, Tamil Nadu, India, **3** Department of General Medicine, Royal Adelaide hospital, Adelaide, Australia, **4** Department of Biostatistics, Christian Medical College Vellore, Vellore, Tamil Nadu, India

☯ These authors contributed equally to this work.
‡ MK, TVP and BY also contributed equally to this work.
* george.thatsme@gmail.com

**Data Availability Statement:** All relevant data are within the manuscript and its Supporting Information files.

## Abstract

### Introduction

Current diagnostic methods used in Central Nervous System Tuberculosis (CNS TB) are limited by the paucibacillary nature of this form of tuberculosis. Posterior pituitary bright spot (PPBS) refers to an area of T1 hyperintensity in the posterior pituitary in MR imaging of the brain. It is found in 80–90% of healthy children and adults. In children with CNS TB, nearly half have absence of PPBS. This finding has not been described in adults. Our study looked for absence of PPBS in MR imaging and its association with CNS tuberculosis.

### Objective

To study prevalence of the absence of PPBS in patients with CNS tuberculosis when compared to a control group of normal patients.

### Methods

This was a retrospective case-control study of 100 patients with CNS tuberculosis and 200 controls (matched in 1:2 ratio) of patients with normal MRI brain. The MRI images were presented to a blinded radiologist in a randomised sequence to report for absence of PPBS. The data was subsequently analysed to look for association of absence of PPBS with CNS tuberculosis.

### Results

Absence of PPBS (cases (47%), controls (8.5%)) was significantly associated with CNS tuberculosis in (Odds ratio-7.90, 95%CI 4.04–15.44, P-value<0.0001). The specificity, sensitivity, positive predictive value and positive likelihood ratio are 91.5%, 47%, 73.4% and

**Funding:** This study received assistance in funding from the Vellore CMC Foundation and Office of Research, CMC Vellore. The funders had no role in study design, data collection and analysis, decision to publish, or preparation of the manuscript.

**Competing interests:** The authors have declared that no competing interests exist.

5.53 respectively. Adding of absence of PPBS as an additional radiological feature in diagnosis of CNS TB increased the sensitivity from 77% to 84%.

## Conclusion

Absence of PPBS is significantly associated with CNS tuberculosis and could be a relatively simple diagnostic aid in the diagnosis of CNS tuberculosis.

## Introduction

CNS tuberculosis (CNS TB), the most severe form of tuberculosis, is a leading cause of neurological infections in the world, especially in TB endemic areas like south Asia [1, 2]. CNS TB has remained to be a diagnostic challenge in medicine since 17th century when it was first described by Willis [3].

Despite advances in microbiological isolation, only one in three patients diagnosed as CNS TB have definite CNS TB [4]. With advent of newer MR imaging modalities, presence of radiological features like basal exudates, meningeal enhancement, tuberculoma, infarcts and tuberculoma has helped physicians in dealing with cases where microbiological evidence is lacking [5].

In 80–90% of healthy children and adults, there is an area of T1 hyperintensity in the posterior pituitary on T1 weighted mid sagittal MRI images described as the 'Bright Spot' [6, 7]. This is thought to result from the T1-shortening effect of stored vasopressin in the posterior lobe of the pituitary [8]. An enlarged pituitary bright spot is seen in certain physiological conditions such as newborn, pregnancy or lactation but is usually anterior in position [9]. The loss of normal posterior pituitary bright spot (PPBS) was previously described in primary diabetes insipidus and in water intoxication [10, 11]. Andronikou et al had found that there was absence of PPBS in 55% of children with TB meningitis [12]. However, its absence in adults with CNS tuberculosis hasn't been studied so far.

Hence, we planned a case-control study to look at increase in prevalence of the radiological feature of absence of PPBS in MR imaging and its association with CNS TB in adults.

## Methods

We conducted a retrospective case-control study using medical records of patients treated between January 2014 to April 2019 at a tertiary care hospital in South India. The need for informed consent was waived and this study was approved by the Institutional Research Board and Ethics Committee of the Institution (IRB Min.No.12031).

All patients with TB involving the brain (CNS TB) as diagnosed by treating physician were included during this time period as cases. Two controls were recruited for each case. From electronic medical records, patients who had MRI brain under the same unit in the same time period, and whose MRI was reported as normal were identified. Using computer generated random numbers 200 controls were selected.

The MRI images of both cases and controls were presented in a random sequence, after removing patient identifiers, to a blinded consultant radiologist to report the radiological feature of absence of PPBS.

The PPBS was identified on sagittal T1-weighted images (Fig 1). The image slice on which the PPBS was the largest was selected for measurements. PPBS was identified to be present

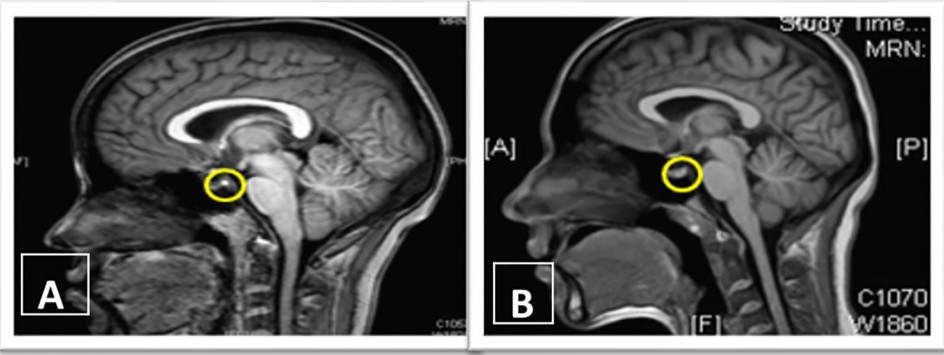

**Fig 1. Posterior Pituitary Bright Spot (PPBS).** (A) Presence of PPBS and(B) Absence of PPBS.

(normal) if it measured between 1.2 and 8.5 mm in the long axis and between 0.4 and 4.4 mm in the short axis in patients who do not have any pituitary abnormality [8].

Three common clinical score for CNS TB were used for grading of cases–modified MRC score, Thwaites diagnostic index score and Lancet consensus score [13–15].

Summary data was presented as mean (standard deviation, SD) or median with interquartile range for continuous variables and categorical variables as numbers and percentages. The characteristics of cases of CNS TB or control (normal MRI brain) were compared using a t-test for continuous data and categorical data was compared using Chi-square/Fisher's exact test as appropriate. Adjusted analysis with important factors associated with cases and controls were explored using logistic regression analysis and expressed as Odds Ratio (OR) with 95% Confidence Intervals (CI). Statistical significance was defined as P<0.05. The data was entered using Epidata v3.1 and analyses were performed using SPSS version 25.

## Results

One-hundred cases and 200 controls were recruited as shown in Fig 2.

Among cases fever (95%) and headache (84%) were the most common clinical symptoms while neck stiffness (69%) was the common clinical sign. (Table 1) cases (47%) compared to controls (8.5%) CSF examination was safe to perform in 93/100 cases among whom 15 /93 (16.13%) were positive for tuberculosis on mycobacterial growth indicator tube (MGIT). Seventy percent were MRC stage 2 or 3 and 16% had a definitive diagnosis on Lancet consensus score.

Compared to the controls, cases had higher mean age at presentation (37.92 ±15.62 vs. 32.48 ± 6.98), more males (60% vs. 42%), more diabetics (17% vs. 5.5%) and higher proportion with past history of tuberculosis (16% vs. 3%) (Table 2).

PPBS was absent in 47% (n = 47) of the cases when compared to 8.5% (n = 17) of the controls which was statistically significant (Adjusted OR 7.90(95%CI-4.04–15.44). The specificity of "absence of PPBS" in CNS TB is 91.5% (95%CI- 86.7–95), sensitivity is 47% (95%CI- 36.9–57.2) and positive predictive value is 73.4% (95%CI- 62.6–82) and positive likelihood ratio is 5.53 (95%CI- 3.35–9.12).

The "absence of PPBS" was compared with other characteristic radiological features of CNS TB like basal exudates/meningeal enhancement (BM), arachnoiditis (A), endarteritis (E), tuberculoma (T) and hydrocephalus (H). The absence of PPBS (47%) was the second most common feature after basal exudates and meningeal enhancement (60%) and seen more commonly than hydrocephalus (32%), tuberculoma (24%), endarteritis (22%) and arachnoiditis (8%) (Fig 3).

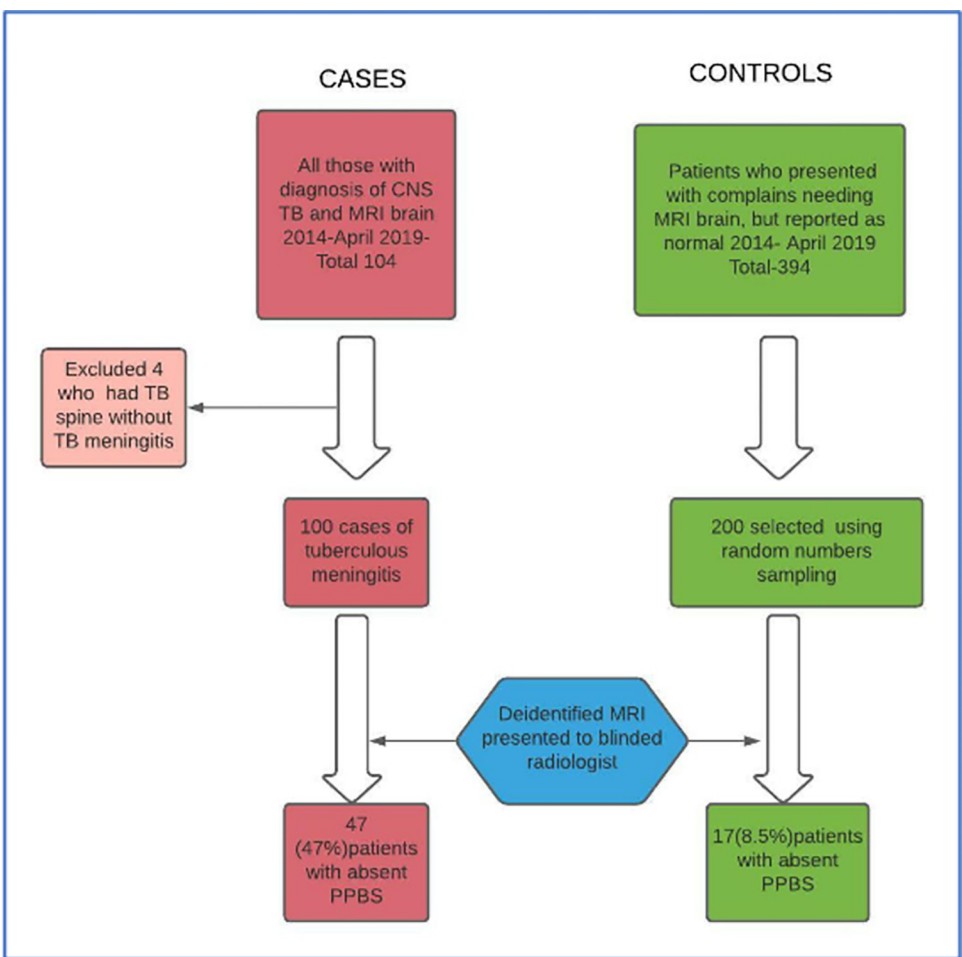

**Fig 2. STROBE flow diagram.**

**Table 1. Characteristics of cases (CNS TB).**

| Variable | Cases (CNS TB) n = 100 |
|---|---|
| Duration of symptoms - median (IQR) | 24.5 days(75) |
| CSF–Gram stain | All negative |
| CSF–AFB smear | All negative |
| CSF–MGIT positive | 15/93(16.13%) |
| SCORES | |
| Modified MRC (n = 100) | |
| • Grade 1 | 30 (30%) |
| • Grade 2 | 47 (47%) |
| • Grade 3 | 23 (23%) |
| Thwaites diagnostic index score (n = 93) | |
| - ≤ 4 | 92 (98.9%) |
| - > 4 | 1 (1.1%) |
| LANCET consensus score (n = 93) | |
| • Possible TBM—n (%) | 41 (44.1%) |
| • Probable TBM - n (%) | 37 (39.8%) |
| • Definite TBM—n (%) | 15 (16.1%) |

**Table 2. Baseline characteristics of the cases and controls.**

| Variables | Unadjusted analysis | | | Adjusted analysis* | |
|---|---|---|---|---|---|
| | Cases(CNS TB) (n = 100) | Control (n = 200) | p-value | OR (95% CI) | p-value |
| Age in years (Mean ± SD) | 37.92 ± 15.62 | 32.48 ± 6.98 | <0.001 | 1.03(0.99–1.06) | 0.06 |
| Gender (males) | 60 (60%) | 84 (42%) | 0.005 | 1.72(0.98–3.02) | 0.06 |
| Diabetes mellitus | 17 (17%) | 11 (5.5%) | 0.002 | 1.07(0.37–3.09) | 0.90 |
| Hypertension | 12 (12%) | 28(14%) | 0.76 | | |
| Obstructive airway disease | 3 (3%) | 3 (1.5%) | 0.66 | | |
| Chronic kidney disease | 0 (0%) | 1 (0.5%) | - | | |
| HIV - Seropositive | 5 (5%) | 0 (0%) | - | | |
| Immunosuppression | 1 (1%) | 6 (3%) | 0.50 | | |
| Past history of Tuberculosis | 16 (16%) | 6 (3%) | <0.001 | 4.92(1.68–14.39) | 0.004 |
| Absence of PPBS | 47(47%) | 17 (8.5%) | <0.001 | 7.90(4.04–15.44) | <0.001 |

*Adjusted for Age, gender, Diabetes Mellitus, Past history of TB and Absence of PPBS

Adding of "absence of PPBS" as an additional radiological feature in diagnosis of CNS TB increased the sensitivity from 77% to 84% (Tables 3 and 4).

## Discussion

This is the first study, to our knowledge, assessing prevalence of the radiological feature of absence of PPBS in adults with CNS TB and the added diagnostic value of this feature. Most were MRC grade 2 or 3 at presentation and only 16.1% of them had definite microbiological evidence of TBM.

In our study, the odds of not having PPBS in TBM were 7.90. As a diagnostic test, absence of PPBS had a 91.5% specificity and positive likelihood ratio of 5.53. Adding of absence of

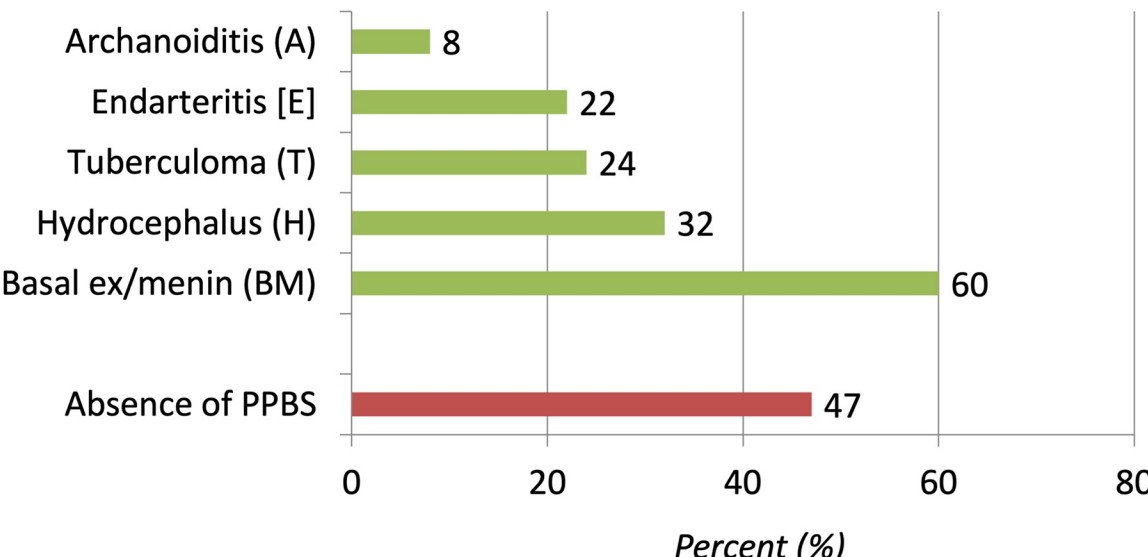

**Fig 3. Prevalence of various radiological features of CNS TB cases.** (BM)Basal exudates or Meningeal enhancement,(A)Arachnoiditis, (E)Endarteritis,(T) Tuberculoma,(H) Hydrocephalus and absence of posterior pituitary bright spot(PPBS).

**Table 3. Sensitivity and specificity of MRI with A/E/T/H/BM for TBM.**

| Any 1 of the 5 MRI features * (A/E/T/H/BM) | TBM cases (N = 100) | Controls (N = 200) | Total Samples (N = 300) | Positive Predictive Value | Negative Predictive Value | Sensitivity | Specificity |
|---|---|---|---|---|---|---|---|
| | number | | | percent | | | |
| Present | 77 | 0 | 77 | 77/77 (100) | | 77/100 (77) | |
| Absent | 23 | 200 | 223 | | 200/223 (89.7) | | 200/200 (100) |

*Any 1 of these 5 MRI features–(A)Archanoiditis, (E)Endarteritis, (T) Tuberculoma,(H) Hydrocephalus, (BM) Basal exudates/Meningeal.

**Table 4. Sensitivity and specificity of MRI for TBM with addition of "Absent PPBS" to A/E/T/H/BM.**

| Any 1 of the 6 MRI features # (A/E/T/H/BM/Absent PPBS) | TBM cases (N = 100) | Controls (N = 200) | Total Samples (N = 300) | Positive Predictive Value | Negative Predictive Value | Sensitivity | Specificity |
|---|---|---|---|---|---|---|---|
| | number | | | percent | | | |
| Present | 84 | 0 | 84 | 84/84 (100) | | 84/100 (84) | |
| Absent | 16 | 200 | 216 | | 200/216 (92.6) | | 200/200 (100) |

# Any 1 of these 6 MRI features–(A)Archanoiditis,(E)Endarteritis, (T) Tuberculoma,(H) Hydrocephalus,(BM) Basal exudates/Meningeal, Absent PPBS.

PPBS as an additional radiological feature in diagnosis of CNS TB increased the diagnostic yield from 77% to 84%, hence highlighting its importance as an additional diagnostic aid.

The pathophysiological basis for absence of PPBS in CNS TB remains to be investigated. The inflammatory response and granulation tissue formation in CNS TB is commonly concentrated around the basal cisterns [16]. CNS TB associated endarteritis and vascular thrombosis could result in destruction of parts of pituitary gland and disruption of the hypothalamic-hypophysial pathways which lie in close anatomical proximity [12]. As a result of these, there could be a decreased storage of vasopressin in posterior pituitary thereby causing absence of PPBS.

In comparison to other radiological features, absence of PPBS is a relatively simple radiological sign which can be easily picked up by a clinician without much radiological background knowledge or training especially in areas where radiology reporting is delayed. (Fig 1).

## Limitations

Though the finding of absence of PPBS was significantly higher in our TBM cases, its robustness in aiding diagnosis will need further prospective studies with controls with non tubercular CNS infections.

## Conclusion

Among adults with suspected CNS tuberculosis this is the first study to show the odds of absence of PPBS was 7.90 in favour of a diagnosis of TB. Absence of PPBS can be a relatively simple radiological aid in diagnosis of adults with suspected CNS tuberculosis.

## Supporting information

**S1 Data. Minimal data sheet.**
(XLSX)

## Author Contributions

**Conceptualization:** Smitesh G. G., Pavithra Mannam, Vignesh Kumar, Tina George, Murugabharathy K., Turaka Vijay Prakash, Bijesh Yadav, Thambu David Sudarsanam.

**Data curation:** Smitesh G. G., Pavithra Mannam, Tina George, Murugabharathy K.

**Formal analysis:** Smitesh G. G., Vignesh Kumar, Bijesh Yadav, Thambu David Sudarsanam.

**Investigation:** Smitesh G. G., Tina George, Thambu David Sudarsanam.

**Methodology:** Smitesh G. G., Pavithra Mannam, Vignesh Kumar, Tina George, Murugabharathy K., Turaka Vijay Prakash, Thambu David Sudarsanam.

**Project administration:** Tina George, Turaka Vijay Prakash, Thambu David Sudarsanam.

**Resources:** Pavithra Mannam, Tina George, Thambu David Sudarsanam.

**Supervision:** Tina George.

**Visualization:** Tina George.

**Writing – original draft:** Smitesh G. G., Pavithra Mannam, Tina George, Murugabharathy K., Thambu David Sudarsanam.

**Writing – review & editing:** Smitesh G. G., Pavithra Mannam, Vignesh Kumar, Tina George, Murugabharathy K., Turaka Vijay Prakash, Bijesh Yadav, Thambu David Sudarsanam.

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
