## [Decision Letter · Decision Letter 0]

4 Jul 2022

PONE-D-22-00965Absence of Posterior Pituitary Bright Spot in adults with CNS Tuberculosis: A Case –Control Study.PLOS ONE

Dear Dr. George,

Thank you for submitting your manuscript to PLOS ONE. After careful consideration, we feel that it has merit but does not fully meet PLOS ONE’s publication criteria as it currently stands. Therefore, we invite you to submit a revised version of the manuscript that addresses the points raised during the review process. Please submit your revised manuscript by Aug 18 2022 11:59PM. If you will need more time than this to complete your revisions, please reply to this message or contact the journal office at plosone@plos.org. Please include the following items when submitting your revised manuscript:A rebuttal letter that responds to each point raised by the academic editor and reviewer(s). You should upload this letter as a separate file labeled 'Response to Reviewers'.A marked-up copy of your manuscript that highlights changes made to the original version. You should upload this as a separate file labeled 'Revised Manuscript with Track Changes'.An unmarked version of your revised paper without tracked changes. You should upload this as a separate file labeled 'Manuscript'.If applicable, we recommend that you deposit your laboratory protocols in protocols.io to enhance the reproducibility of your results. Protocols.io assigns your protocol its own identifier (DOI) so that it can be cited independently in the future. For instructions see: https://journals.plos.org/plosone/s/submission-guidelines#loc-laboratory-protocols. Additionally, PLOS ONE offers an option for publishing peer-reviewed Lab Protocol articles, which describe protocols hosted on protocols.io. Read more information on sharing protocols at https://plos.org/protocols?utm_medium=editorial-email&utm_source=authorletters&utm_campaign=protocols.

We look forward to receiving your revised manuscript.

Kind regards,

Mao-Shui Wang

Academic Editor

PLOS ONE

Journal Requirements:

Reviewers' comments:

Reviewer's Responses to Questions

**Comments to the Author**

1. Is the manuscript technically sound, and do the data support the conclusions?

Reviewer #1: Yes

Reviewer #2: Yes

2. Has the statistical analysis been performed appropriately and rigorously? 

Reviewer #1: Yes

Reviewer #2: Yes

3. Have the authors made all data underlying the findings in their manuscript fully available?

Reviewer #1: Yes

Reviewer #2: Yes

4. Is the manuscript presented in an intelligible fashion and written in standard English?

Reviewer #1: Yes

Reviewer #2: Yes

5. Review Comments to the Author

Reviewer #1: In this manuscript titled"Absence of Posterior Pituitary Bright Spot in adults with CNS Tuberculosis: A Case –Control Study" The authors compared the MRI of two large cohorts patients with CNS tuberculosis and patients controls normal. I have no specific comment. The quality of the MRI and the absence of PPBS in 47% versus 8.5% patients with CNS tuberculosis compared to the controls are convincing.

Reviewer #2: Its a very good manuscript bridging the gap in knowledge. There are few spacing issues that could be corrected. Rest looks OK. One reference no 14 should be in sentence case after first capital letter.

6. PLOS authors have the option to publish the peer review history of their article (what does this mean?). If published, this will include your full peer review and any attached files.

Reviewer #1: No

Reviewer #2: **Yes: **SANKALP YADAV

---

## [Author Response · Author response to Decision Letter 0]

1 Sep 2022

To 

The Academic Editor

PLOS ONE

And Reviewers

Subject- Corrections to manuscript number PONE-D-22-00965

Dear Sir,

I thank you and all the reviewers for your time and effort in reviewing our paper.

Thank you for all the positive reviews and the suggested changes. Please find my response the same below.

Response to changes suggested by the editor

1- I have herewith submitted a marked up copy with highlights made to the changes and an unmarked version of the same.

2- The manuscript has been changed to fit the PLOS ONEs style requirements.

3- The study’s minimal data set has been uploaded to the supporting information files .

4- References have been revised as recommended.

5- I have also changed the affiliation of one of the co authors Dr Vijay Prakash Turaka as he has recently changed his affiliation.

Response to reviewer 2 section 5 

 Thank you for your comments and the changes as suggested by you have been made.

I would again like to thank the editorial team and reviewers for all the feedback and support provided to our paper. We hope for a positive response.

Thanking you

Dr Tina George

Associate Professor

Department of Medicine 

Christian Medical college Vellore

Tamil Nadu- 632004

India

Email- george.thastme@gmail.com, tinageorge@cmcvellore.ac.in

---

## [Editor Report · Decision Letter 1]

19 Sep 2022

Absence of Posterior Pituitary Bright Spot in adults with CNS Tuberculosis: A Case –Control Study.

PONE-D-22-00965R1

Dear Dr. George,

We’re pleased to inform you that your manuscript has been judged scientifically suitable for publication and will be formally accepted for publication once it meets all outstanding technical requirements.

Kind regards,

Mao-Shui Wang

Academic Editor

PLOS ONE
---

## [Editor Report · Acceptance letter]

23 Sep 2022

PONE-D-22-00965R1 

Absence of Posterior Pituitary Bright Spot in adults with CNS Tuberculosis: A Case –Control Study. 

Dear Dr. George:

I'm pleased to inform you that your manuscript has been deemed suitable for publication in PLOS ONE. Congratulations! Your manuscript is now with our production department. 

Kind regards, 

on behalf of

Dr. Mao-Shui Wang 

Academic Editor

PLOS ONE